



# Improved retrieval of SO$_2$ plume height from TROPOMI using an iterative Covariance-Based Retrieval Algorithm

**Nicolas Theys[1], Christophe Lerot[1], Hugues Brenot[1], Jeroen van Gent[1], Isabelle De Smedt[1], Lieven Clarisse[2], Mike Burton[3], Matthew Varnam[3], Michel Van Roozendael[1]**

[1] Royal Belgian Institute for Space Aeronomy (BIRA-IASB), Brussels, Belgium.

[2] Université libre de Bruxelles (ULB), Spectroscopy, Quantum Chemistry and Atmospheric Remote Sensing (SQUARES), C. P. 160/09, Brussels, Belgium.

[3] School of Earth and Environmental Sciences, University of Manchester, Oxford Road, Manchester, M139PL, UK

*Correspondence to:* N. Theys (theys@aeronomie.be )

## ABSTRACT

Knowledge of sulfur dioxide layer height (SO$_2$ LH) is important to understand volcanic eruption processes, the climate impact of SO$_2$ emissions and to mitigate volcanic risk for civil aviation. However, the estimation of SO$_2$ LH from ground-based instruments is challenging in particular for rapidly evolving and sustained eruptions. Satellite wide-swath nadir observations have the advantage to cover large-scale plumes and the potential to provide key information on SO$_2$ LH. In the ultraviolet, SO$_2$ LH retrievals leverage the fact that, for large SO$_2$ columns, the light path and its associated air mass factor (AMF) depends on the SO$_2$ absorption (and therefore on the vertical distribution of SO$_2$), and SO$_2$ LH information can be obtained from the analysis of measured back-scattered radiances coupled with radiative transfer simulations. However, existing algorithms are mainly sensitive to SO$_2$ LH for SO$_2$ vertical columns of at least 20 DU. Here we develop a new SO$_2$ LH algorithm and apply it to observations from the high spatial resolution TROPOspheric Monitoring Instrument (TROPOMI). It is based on an SO$_2$ optical depth look-up-table and an iterative approach. The strength of this scheme lies in the fact that it is a Covariance-Based Retrieval Algorithm (COBRA; Theys et al., 2021). This means that the SO$_2$-free contribution of the measured optical depth is treated in an optimal way, resulting in an improvement of the SO$_2$





LH sensitivity to $SO_2$ columns as low as 5 DU, with a precision better than 2km. We demonstrate
the value of this new data through a number of examples and comparison with satellite plume
height estimates (from IASI and CALIOP), and back trajectory analyses. The comparisons
indicates an $SO_2$ LH accuracy of 1-2 km, expect for some difficult observation conditions.
**1. INTRODUCTION**
Volcanic eruptions can emit large quantities of rock fragments and fine particles (ash) into the
atmosphere as well as several trace gases, such as carbon dioxide ($CO_2$), sulphur species ($SO_2$,
$H_2S$), halogens (HCl, HBr, HF), and water vapour. These volcanic ejecta can have a tremendous
impact on human health, society and nature, and on air traffic safety. In particular, emission of
sulphur dioxide ($SO_2$) receives considerable attention due to its subsequent conversion into
aerosols and potentially strong effect on global climate (Robock, 2000). Among the emitted
constituents, $SO_2$ is also the easiest to detect from ultraviolet (UV) and thermal infrared (TIR)
remote-sensing techniques, and is being used for many decades to monitor volcanoes worldwide.
In order to understand volcanic processes and assess the impact of eruptions, it is crucial to
measure not only the total abundance of $SO_2$ but also the height of the $SO_2$ plume. This information
is important for (1) aviation actors such as Volcanic Ash Advisory Centres (VAACs) in case ash
and $SO_2$ clouds are collocated; sulfur alone is also becoming increasingly recognized as causing
long-term damage to aircraft engines (mainly because of sulfuric acid), (2) volcanology as it
informs on eruption rate, eruption type and underlying volcanic processes (e.g., Mastin et al.,
2009), (3) atmospheric chemistry and climate research, e.g. to model the impact of volcanic
eruptions on air quality (Schmidt et al., 2015) or to study the partly understood role of modest
volcanic eruptions on climate forcing (Solomon et al., 2011; Vernier et al., 2011; Santer et al.,
2014), and (4) the estimation of $SO_2$ emissions, as the measured $SO_2$ abundances are often directly
dependent on the knowledge of the $SO_2$ vertical distribution.
Ground-based cameras can be used to routinely monitor plume heights (e.g., Scollo et al., 2014)
but these measurements are performed very near-field. For large and sustained volcanic eruptions,
estimation of plume heights is very difficult in practice – not to say impossible – and the available
measurements generally suffer from poor or infrequent sampling of the volcanic plumes.
Moreover, many volcanoes on the globe are not monitored. Consequently, satellite nadir sensors
with large swaths and frequent revisiting time offer the best solution to cover completely the



emitted volcanic cloud.
Space nadir sensors have provided global measurements of $SO_2$ vertical columns and masses for
more than 40 years (Carn et al., 2016, and references therein). However, the retrieval of $SO_2$ plume
height (also referred to $SO_2$ layer height, $SO_2$ LH) from satellite hyperspectral measurements is a
relatively recent development. In the TIR, global $SO_2$ LH retrievals from the Infrared Atmospheric
Sounding Interferometer (IASI) by Carboni et al. (2012) and Clarisse et al. (2014) and the Cross-
track Infrared Sounder (CrIS) by Hyman and Pavolonis (2020) proved to have an excellent
sensitivity to the $SO_2$ height above ~5 km, even for $SO_2$ columns at 1 DU level (Dobson unit – 1
DU: $2.69 \times 10^{16}$ molecules $cm^{-2}$). In the UV spectral range, the sensitivity to $SO_2$ is better at lower
altitudes and the first studies using full radiative transfer calculation schemes were from Yang et
al. (2010) and Nowlan et al. (2011), based on the Ozone Monitoring Instrument (OMI; Levelt et
al., 2006) and the Global Ozone Monitoring Experiment-2 (GOME-2; Munro et al., 2006). More
recently, new approaches based on Inverse Learning Machine schemes have become available for
GOME-2 (Efremenko et al., 2017), TROPOspheric Monitoring Instrument-TROPOMI (Hedelt et
al., 2019) and OMI (Fedkin et al., 2021). These algorithms greatly improve the computational
performance of the previously published techniques. However, all the UV schemes referenced
above have demonstrated sensitivity to $SO_2$ LH only for $SO_2$ vertical columns of more than ~ 20
DU, which limits their use to relatively large volcanic events. In this paper, we present a new UV
spectral fitting algorithm allowing to retrieve $SO_2$ LH for $SO_2$ columns as low as 5 DU, and for
$SO_2$ layer heights as low as 1km. This scheme is an extension of our recently published $SO_2$
Covariance-Based Retrieval Algorithm (COBRA; Theys et al., 2021) that enables drastic reduction
in spectral interferences and retrieval noise. Here we combine COBRA with an iterative look-up
table approach to treat the non-linear $SO_2$ contribution to the measured signal. This allows joint
retrieval of the $SO_2$ vertical column density (VCD) and $SO_2$ layer height with improved sensitivity
while avoiding time-consuming on-line radiative transfer simulations. We apply this technique to
measurements from TROPOMI (Veefkind et al., 2012) aboard the Sentinel-5 Precursor (S-5P)
satellite. The motivation is obviously its high spatial resolution of $3.5 \times 5.5$ km². TROPOMI
resolves locally enhanced $SO_2$ columns much better than predecessor instruments like OMI (Theys
et al., 2019). The retrieval of $SO_2$ LH is therefore expected to be possible for several degassing
volcanoes. This has the potential to enhance our capability of monitoring height-resolved volcanic
plumes globally in the troposphere. In addition, for strong eruptions, retrieved $SO_2$ LH (and $SO_2$



vertical columns) at high spatial resolution can also provide unique insights into volcanic
processes, atmosphere-plume interactions and transport (Pardini et al., 2018, references).
The paper is structured as follows. Section 2 describes the algorithm in detail and demonstrates
the performance of the $SO_2$ LH retrieval. In Section 3, the results are evaluated against other
satellite data sets and dispersion model results. Conclusions and perspectives are given in Section

6 4.

## 8   2.   ALGORITHM DESCRIPTION

The theoretical basis for a joint retrieval of $SO_2$ column amount and layer altitude from satellite
nadir back-scattered UV measurements is described by Yang et al. (2010) and Nowlan et al.
(2011). The TROPOMI $SO_2$ layer height algorithm, outlined in this section, is an iterative retrieval
scheme. It is conceptually close to these pioneering algorithm studies in the way the $SO_2$ absorption
is handled but differs in the treatment of the other contributions to the measured signal.
We first define the measured top-of-atmosphere total optical depth (OD) by:

$$y_{meas} = y_{SO2} + y_{bckg} + \varepsilon \tag{1}$$

All terms of the equation depend on wavelength (not labelled here, for simplicity). $y_{meas} =$
$-\log(I/I_0)$ is the logarithmic ratio of the wavelength calibrated measured radiance ($I$) and
irradiance ($I_o$) over a given wavelength range, $y_{SO2}$ is the unknown $SO_2$ optical depth, $y_{bckg}$ is the
"background" optical depth, accounting for all contributions to the total OD except that of $SO_2$,
and $\varepsilon$ is the measurement error.
In case of strong $SO_2$ absorption, the optical depth $y_{SO2}$ is fundamentally a non-linear function of
the VCD and LH of $SO_2$, and solving Eq. (1) is non-trivial. However, we assume here that the
expression can be linearized using a Taylor expansion:

$$y_{meas} - y_{SO2,i} \approx \Delta VCD \frac{\partial y_{SO2,i}}{\partial VCD} + \Delta LH \frac{\partial y_{SO2,i}}{\partial LH} + y_{bckg} + \varepsilon \tag{2}$$

$y_{SO2,i}$ is the $SO_2$ OD at the linearization point $y_{SO2,i} = y_{SO2}(VCD_i, LH_i)$, $\partial y_{SO2,i}/\partial VCD$ and
$\partial y_{SO2,i}/\partial LH$ are the corresponding partial derivatives with respect to the $SO_2$ VCD and LH





(Jacobians), and $\Delta VCD$ and $\Delta LH$ are the VCD and LH increments. Index i stands for the i[th]
iteration.
To solve Eq. (2), we developed a hybrid method. To model the $SO_2$ signal, the algorithm makes
use of a large look-up-table (LUT) of $SO_2$ OD. At each iteration, improved estimations of VCD
and LH become available. These results are used to update the $SO_2$ OD and Jacobians for the next
calculation, until convergence is reached. This part of the algorithm will be detailed in section 2.1.
To treat the background and error terms of Eq. (2), we propose a COBRA method. In brief, instead
of fitting the background optical depth, the algorithm considers a representative set of measured
spectra uncontaminated by $SO_2$, and characterized by a mean optical depth $\bar{y}$ and a covariance
matrix $S$, to represent statistically $y_{bckg} + \varepsilon$. The idea of the method, initially presented by Walker
et al. (2011) and further developed in other studies (e.g., Carboni et al., 2012; Clarisse et al., 2014;
Theys et al., 2021), is to consider $y_{bckg} + \varepsilon$ as an error term, and to interpret $S$ as a generalized
error covariance matrix. Thus, the solution of the inverse problem can be expressed (Rodgers,

14   2000):

$$\hat{x}_{i+1} = \hat{x}_i + (k_i^T S^{-1} k_i)^{-1} k_i^T S^{-1} (y_{meas} - y_{SO2,i} - \bar{y}) \qquad (3)$$
where $\hat{x}_i$ is the retrieved state vector ($[LH_i, VCD_i]^T$) and $k_i$ is the $SO_2$ forward model
($[\partial y_{SO2,i}/\partial LH \ \ \partial y_{SO2,i}/\partial VCD]$).
It should be emphasized that the matrix $S$ accounts (if well-constructed) for most atmospheric
background and instrumental-related variability of the spectra, including cross-correlations. The
strength of the algorithm lies in the fact that only two parameters are retrieved ($SO_2$ LH and VCD).
As will be shown in sections 2.3 and 3.1, this constitutes a significant advance in terms of retrieval
sensitivity compared to a classical fitting approach such as the Differential Optical Absorption
Spectroscopy (DOAS; Platt and Stutz, 2002), where multiple parameters are fitted in addition to
$SO_2$ LH and VCD.
The following two sections describe in more detail the $SO_2$ LUT approach and the specific
algorithm settings.



## 2.1 $SO_2$ optical depth look-up-table: description

Forward modeled $SO_2$ spectra are based on the LInearized Discrete Ordinate Radiative Transfer (LIDORT) model version RRS 2.2 (Spurr et al., 2008). The input data used to set the atmosphere and spectroscopy are detailed in Table A1. Simulations were carried out to cover a large range of possible measurement conditions, using different combinations of LUT entries for the observation geometry, total ozone column, Lambertian Equivalent Reflectivity (LER) and $SO_2$ vertical profiles (Table 1). More details are given below. The simulation results are the $SO_2$ slant optical depth spectra over a wavelength range from 309 to 329 nm and at a spectral sampling of 0.05 nm. Note that the spectroscopic input data (absorption cross-sections and solar spectrum) were not pre-convolved with the Instrumental Spectral Response Function (ISRF) of TROPOMI but rather with a box-car function of 0.05 nm width. Therefore, the simulations are not instrument-specific. For application to TROPOMI, the $SO_2$ OD spectra were convolved with the ISRF, and a specific correction for the so-called solar-$I_0$ effect (Aliwell et al., 2002) was applied, as it turned out to be important for large $SO_2$ VCDs (100-1000 DU). It should be noted that the TROPOMI ISRF parameters vary smoothly with the position across-track (450 in total). For practical ease of use, we assumed that the across-track dependence is well encapsulated by the viewing zenith angle (VZA) entry. To represent the full swath, we used the sign convention of negative/positive VZA for W/E, respectively. For each of the VZA grid point (Table 1), a slit function of the TROPOMI detector column was associated with the closest mean VZA. This appears to be a good approximation for TROPOMI and avoids having 450 different LUTs.

From the LUT of $SO_2$ OD spectra, the algorithm extracts a sub-LUT for a given TROPOMI measurement by linear interpolation. To do so, the observation angles at the ground pixel location are used. Input on total ozone is obtained from the TROPOMI off-line total ozone column product (Garane et al., 2019). The latter is well suited for the present application, as it is weakly affected by spectral interferences with $SO_2$ (bias of only few % in case of strong eruptions, see discussion in Lerot et al., 2014). In addition to the observation geometry and total ozone absorption, the measurement sensitivity to $SO_2$ is also strongly dependent on the surface reflectance and the presence of clouds or aerosols layer. Here we assume that the radiative transfer in the atmosphere can be sufficiently represented through a lower bound Lambertian Equivalent Reflector. The LER approach works very well in principle when the $SO_2$ plume is above a cloud or aerosol layer.



However, it has limited applicability for cases where $SO_2$ and aerosols are mixed, especially for
highly absorbing aerosols such as volcanic ash. This aspect will be further discussed in Section 3.
The LER is characterized by effective parameters, height and albedo, that are determined for each
pixel. The LER height is computed as the cloud-fraction weighted mean of the cloud and ground
altitudes. Cloud parameters are from the operational cloud product OCRA/ROCINN CRB (Loyola
et al., 2018). For the LER albedo, it is constrained by TROPOMI measured radiance averaged over
339.5-340.5 nm, a range mostly unaffected by trace gas absorption ($O_3$ and $SO_2$). The LER albedo
is retrieved by matching the measured mean radiance to a LUT of radiances (generated in parallel
to the $SO_2$ OD LUT), and which depends on SZA, VZA, RAA, surface height and albedo, with
the same grid definition as in Table 1. The simulated radiances are convolved and averaged over
the same wavelength range as TROPOMI.
Table 1: Physical parameters that define the $SO_2$ optical depth look-up-table. The total number of
spectra is about $38.5 \times 10^6$. In practice, note that LUT interpolation is performed along the cosine
of SZA and VZA.

| Parameter | Grid values | Number of grid points |
|---|---|---|
| Solar zenith angle (SZA) | 10, 20, 30, 40, 50, 60, 70 (°) | 7 |
| Viewing zenith angle (VZA) | -70, -60, -50, -40, -30, -20, -10, 0, 10, 20, 30, 40, 50, 60, 70 (°) | 15 |
| Relative azimuth angle (RAA) | 0, 45, 90, 135, 180 (°) | 5 |
| Total $O_3$ column | 145, 175, 205, 235, 295, 355, 415, 475, 535 (DU) | 9 |
| Albedo | 0, 5, 10, 20, 40, 60, 80, 100 (%) | 8 |
| Surface height | 0, 1, 2, 4, 6, 9 (km) | 6 |
| $SO_2$ column | 1, 2, 5, 10, 25, 50, 100, 200, 500, 1000 (DU) | 10 |
| $SO_2$ height | 1, 2, 3, 4, 5, 6, 7, 8, 9, 10, 11, 12, 13, 14, 15, 20, 25 (km) | 17 |
| Wavelength | 309-329 nm (0.1 nm step, after convolution) | 201 |





From the interpolation step, a sub-LUT of $SO_2$ OD spectra is obtained, which depends only on
$SO_2$ VCD and LH. From this table, the $SO_2$ VCD and LH Jacobians are derived by simple discrete
derivatives. These functions are essential for the retrieval (Eq. 3). Example $SO_2$ LH Jacobians are
presented in Figure 1, for a fixed $SO_2$ VCD of 25 DU and representative LER albedos of 5% (left)
and 80% (right), representative of typical clear-sky and fully cloudy conditions, respectively. For
this example, we observe the largest sensitivity to $SO_2$ LH for low albedo and low $SO_2$ peak height.
This behavior is expected as most of the altitude information comes from the way $SO_2$ alters the
availability of photons to be scattered by air molecules below the $SO_2$ layer (Yang et al., 2010).
Conversely, for high $SO_2$ height or high albedo (e.g., for an underlying cloud), the scattering
weighting functions are weakly dependent on the altitude, and the information on $SO_2$ LH appears
to be less accessible. The performance of the algorithm under various conditions will be discussed
further in Section 2.3.

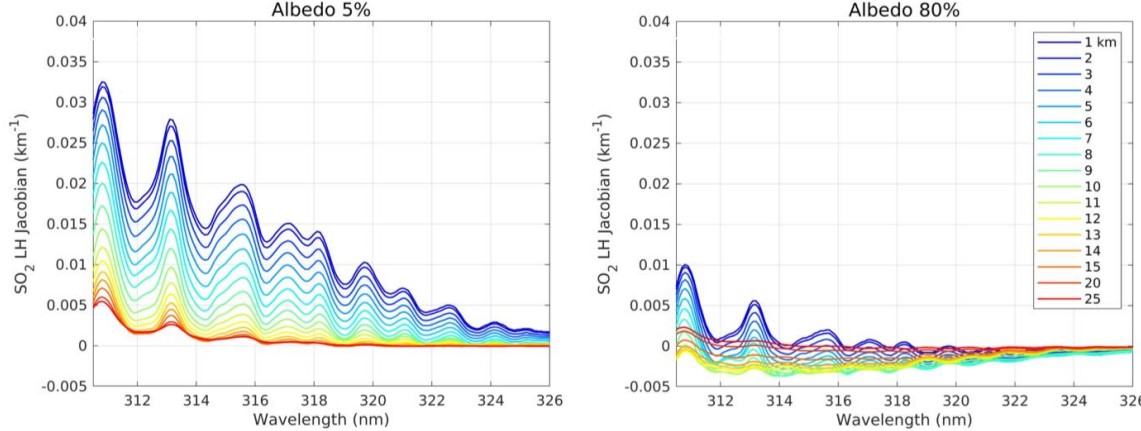

Figure 1: Examples of $SO_2$ optical depth Jacobians with respect to LH for different $SO_2$ peak
heights (1-25 km), and LER albedos of 5% (left) and 80% (right). The spectra correspond to SZA:
30°, VZA: 0°, RAA: 0°, $SO_2$ column: 25 DU, ozone column: 295 DU, surface height: 0 km.
**2.2 LUT-COBRA settings**
The retrieval of $SO_2$ vertical column and height is performed from the analysis of measured
radiances in the spectral range from 310.5 to 326 nm (TROPOMI band 3). The algorithm starts
from an a priori pair ($x_o$) of $SO_2$ VCD and LH. The initial value $VCD_o$ is taken as the output of the





operational TROPOMI $SO_2$ column product for a plume height of 7 km (Theys et al., 2017). The
height $z_o$ is 7 km, except if the LER height is greater than 5km. In that case, $z_o$ is equal to the LER
height + 2km. First, the $SO_2$ optical depth and Jacobians spectra for $x_o$ are derived from the LUT
(as described in section 2.1), and interpolated on the wavelength grid of the measurement. Then
the results of the fit (Eq. 3) are used to calculate new $SO_2$ spectra for the next calculation and the
retrieval is repeated until the inverted LH and VCD do not change from one iteration to the next
by more than 500 m and 10% respectively, or if the number of iterations exceeds a limit value
(fixed to 10). Note that for some iterations, the algorithm gives $SO_2$ LH occasionally outside the
$SO_2$ height grid. For those cases, the $SO_2$ height is forced to the grid minimum height +1 km or
the grid maximum height -1km, depending if the height is below or above the grid
minimum/maximum height respectively. More rarely, the same can happen for the retrieved $SO_2$
VCD and then the $SO_2$ VCD is set to $VCD_o$ for the next iteration.
A key information in the retrieval process is the covariance matrix $S$ (and mean optical depth $\bar{y}$),
as it directly influences the sensitivity of the retrieval (Eq. 3). For the construction of $S$ and $\bar{y}$ we
used a set of measured $SO_2$-free spectra, following an approach analogous to our previous study
(Theys et al., 2021). In brief, for each TROPOMI observation for which the $SO_2$ LH algorithm is
applied (see next section), we consider the spectral data of the corresponding orbit and TROPOMI
row. To represent best the zonal dependence, we select the radiance spectra of 300 pixels along
the flight direction (i.e., ±150 indices along track), and the pixels with observable $SO_2$ amounts
(with VCD>$2.5 \times$ VCD retrieval error) are filtered out. To keep a viable number of spectra for the
covariance calculation (at least 100), we also allow the number of pixels along the flight direction
to increase, if necessary. Note that an upper limit on the SZA is fixed to 65° in order to exclude
difficult conditions with high ozone absorption.
It should be noted that the quality of the LUT-COBRA results depends strongly on the signal of
$SO_2$. This aspect is addressed in the next section.





**2.3 Performance of the retrievals**
Following Rodgers (2000), the estimated error covariance of the solution (Eq. 3) is given by:
$\hat{S}_i = (k_i^T S^{-1} k_i)^{-1}$                                                            (4)
This matrix can be calculated for each TROPOMI pixel and the square root of the diagonal
elements of $\hat{S}_i$ provide error estimates on the retrieved $SO_2$ LH and VCD.
To demonstrate the sensitivity of the algorithm for different $SO_2$ altitudes and vertical columns,
the $SO_2$ LH error was computed for predefined pairs of Jacobians, and a fixed covariance matrix
$S$. The results are summarized in Figure 2 (left) for typical observation conditions in the tropics,
over a dark surface. From this example, it is clear the $SO_2$ LH retrieval uncertainty decreases for
high $SO_2$ columns. This is obvious as the $SO_2$ signal dominates all variability contributions. The
results also suggest that the algorithm performs better for $SO_2$ at low heights. This behavior is
logical, and is in line with the dependence of the $SO_2$ LH Jacobians with the $SO_2$ height (see Fig.
1 left, and related discussion). It is interesting to note that the observed retrieval performance
dependence with $SO_2$ height is complementary to the one found for thermal infrared nadir
sounders, like IASI or CrIS. Indeed, Carboni et al. (2012) and Clarisse et al. (2014) demonstrated
(using IASI) that the best $SO_2$ height retrieval is achieved for $SO_2$ plumes in the upper
troposphere/lower stratosphere (UTLS) while in the lower troposphere (below 3-5 km), the
sensitivity to the $SO_2$ height is strongly reduced, as a result of water vapor absorption. This
complementarity is further addressed in Section 3.
To compare the performance of the LUT-COBRA with more classical fitting approaches, we have
also developed a modified DOAS algorithm, referred as LUT-DOAS in the following. In short,
the forward model matrix was expanded to include not only the LH and VCD Jacobians (grouped
as $k_i$) but also other spectra, used to fit the measured OD. Essentially, we implemented a linearized
version of the DOAS scheme used in the operational TROPOMI $SO_2$ algorithm (Theys et al.,
2017). More precisely, 13 spectral functions are used to model the ozone absorption, Ring effect,
broadband component (in the form of a 3[rd] order polynomial), spectral shift and squeeze (Beirle et
al., 2013), and linear intensity offset. Based on this DOAS-type forward model matrix, the $SO_2$
layer height error for the LUT-DOAS scheme was calculated using Eq. 4, by replacing the
covariance matrix $S$ with an identity matrix, divided by the square of the signal-to-noise ratio





(SNR). The latter was fixed to 800, a typical SNR of TROPOMI radiances over the fitting window
considered. The results are presented in Figure 2 (right).

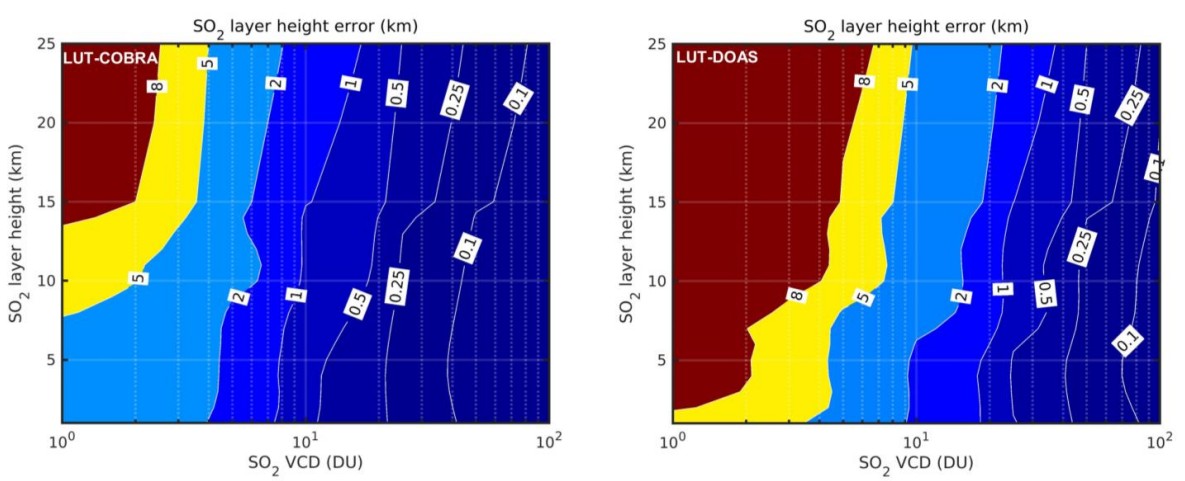

Figure 2: SO$_2$ layer height theoretical uncertainty for (left) LUT-COBRA and (right) LUT-DOAS
schemes. The results correspond to SZA: 30°, VZA: 0°, RAA: 0°, ozone column: 295 DU, surface
height: 0 km, surface albedo: 5%.
This example indicates that, in theory, the LUT-DOAS scheme yields reasonable height retrievals
with uncertainty of 1-2 km for SO$_2$ columns greater than 10-40 DU, depending on the SO$_2$ height.
This finding is mostly consistent with previous studies on UV retrievals of SO$_2$ plume height (e.g.,
Nowlan et al., 2011; Hedelt et al., 2019).
However, compared to the LUT-DOAS algorithm, Figure 2 suggests that our LUT-COBRA
scheme significantly improves the SO$_2$ layer height error by a factor of 2-3. This is an appealing
aspect of the LUT-COBRA approach, as it enables the application of the SO$_2$ LH retrievals to SO$_2$
columns as low as 5 DU.
The performance results of Figure 2 were repeated for other observation conditions. In particular,
a high reflectance scenario (80%) was tested to represent the situation of an SO$_2$ plume lying above
a cloud deck. The results show no significant change in the performance of the retrievals indicating
that underlying clouds have little impact on the sensitivity to SO$_2$ LH. In a way, this is counter-
intuitive when looking at Figure 1, but one should keep in mind that the algorithm retrieves both



LH and VCD of $SO_2$. Clearly, high reflectance conditions help better constrain the $SO_2$ vertical
column (especially for low $SO_2$ heights) which in turn is beneficial to access the spectral
information on $SO_2$ LH. Another interesting result is to test the sensitivity of the retrieval as a
function of the observation angles. For instance, increasing the SZA first leads to improved results,
because the $SO_2$ signal increases as the optical path through the $SO_2$ layer gets longer, but then the
performance quickly deteriorates for high angles due to large absorption by ozone. Those
conditions are, however, discarded by the algorithm SZA cutoff of 65° (section 2.2).
It should be emphasized that the $SO_2$ layer height error presented here does not account for
systematic uncertainties. It is clear that in many circumstances, forward model errors can actually
dominate the total error on $SO_2$ LH. These errors are generally difficult to evaluate and depend on
the prevailing conditions. In Section 3, examples of TROPOMI results will be presented with
specific attention to possible sources of error. We also refer to Yang et al. (2010) and Nowlan et
al. (2011) for a presentation of the various sources of systematic uncertainties.
In practice, the $SO_2$ layer height error (Eq. 4) can be computed for each TROPOMI pixel. This is
useful as it helps diagnose the retrieval quality. In what follows, the retrievals are considered only
for a $SO_2$ layer height error lower than 2.5 km and retrieved VCD of at least 5 DU. In order to
preselect the spectra that potentially fulfill these criteria (and limit the computational effort), the
TROPOMI operational $SO_2$ product was examined. Only measurements with slant column
densities (a quantity independent of $SO_2$ plume height) larger than 2.5 DU were selected and
processed by the $SO_2$ LH algorithm.

## 3. RESULTS

In this section, we present TROPOMI $SO_2$ LH data and evaluate the results against independent plume height estimates from satellites and back-trajectory modelling.

### 3.1 Comparison with satellite plume height estimates

For a selection of eruption events, we performed comparisons with the IASI $SO_2$ height data of Clarisse et al. (2014), readily available in near real-time in the Support to Aviation Control Service (SACS; Brenot et al., 2014, 2021). Another useful dataset to validate the TROPOMI $SO_2$ height is from the Cloud-Aerosol Lidar with Orthogonal Polarization (CALIOP) instrument onboard the Cloud-Aerosol Lidar and Infrared Pathfinder Satellite Observation (CALIPSO). Here we used the 532 nm total backscatter coefficient profiles from the standard CALIOP level-2 v4 product (CAL_LID_L2_05kmAPro-Standard-V4), available from NASA (https://www-calipso.larc.nasa.gov). Although those datasets are very useful, we note that these can only be used to validate TROPOMI $SO_2$ LH in a rather qualitative way because (1) IASI has a different overpass time than TROPOMI, (2) CALIOP has a narrow swath (resulting in limited sampling of what S-5P observes), and measures aerosols rather than $SO_2$. More validation results are shown in sections 3.2 and 3.3.

The first example is for the Sierra Negra volcano (0.83°S, 91.17°W, Ecuador) that erupted on 26 June 2018 at ~20:00 UTC, according to the Global Volcanism Program (volcano.si.edu). Coincidently, TROPOMI passed over the region shortly after the start of the eruption at ~20:12 UTC, and detected a freshly emitted and nearly undispersed $SO_2$ plume with heights of 3-5 km, in good agreement with the S-5P FP_ILM results of Hedelt et al. (2019). On 27 June, the TROPOMI overpass (orbit 03652, approximate time 19:50 UTC) revealed an $SO_2$ plume distributed in multiple layers. Figure 3a shows the results of the LUT-COBRA. The retrieved $SO_2$ heights are as low as 1-2 km near the vent and up to 18 km further downwind. The characteristic pattern of $SO_2$ height levels observed for the different parts of the plume is consistent with the retrievals of CrIS (Hyman and Pavolonis, 2020). Figure 3 also presents results from IASI on 28 June at ~15:15 UTC ($SO_2$ height images for other dates and acquisition times are accessible on the SACS webpage; http://sacs.aeronomie.be). As can be seen, the TROPOMI LUT-COBRA and IASI $SO_2$ LH results agree qualitatively well considering the relatively large time difference of nearly 20 hours between the two sensors. A notable difference though is for the $SO_2$ plume located below 3 km, which is



barely seen in the IASI data. For this particular example, this is partly because the volcano lies
between two orbits but inspection of other $SO_2$ images does not reveal significant $SO_2$ detections
below 3 km by IASI. The reason is likely the limited sensitivity of IASI in the lowermost
troposphere, particularly in the tropics.
In addition to the LUT-COBRA results, Figure 3b also presents the corresponding retrievals from
the LUT-DOAS implementation (introduced in Section 2.3). Here we show the results for retrieved
$SO_2$ VCDs greater than 20 DU in order to keep the $SO_2$ height data with retrieval errors better than
1- 2 km (Figure 2, right panel). This threshold is the same as in Hedelt et al. (2019). Overall, the
$SO_2$ LHs from LUT-DOAS are in close agreement with the LUT-COBRA results, for the pixels
in common. The LUT-DOAS values also match very well the results of S-5P FP_ILM, Fig. 10b
of Hedelt et al. (2019). However, this example of Figure 3 clearly demonstrates that the LUT-
COBRA is able to retrieve $SO_2$ LH for many more pixels with greater sensitivity than the LUT-
DOAS approach. In the following, only the LUT-COBRA results will be presented and discussed.
A second illustration of the LUT-COBRA $SO_2$ LH results is for the Raikoke volcano (48.29°N,
153.25°E, Kuril Islands, Russia) that erupted on 21 June 2019 with multiple explosions that started
at ~18:00 UTC and lasted several hours. The eruption emitted enormous amounts of $SO_2$ in the
atmosphere, around 1.5 Tg (e.g., de Leeuw et al., 2021), as well as volcanic ash. Raikoke is
therefore a good case to test the $SO_2$ LH algorithm under extreme conditions. Note that the eruption
of Raikoke is also well documented and is the subject of an Atmos. Chem. Phys./Atmos. Meas.
Tech./Geosci. Model Dev. special issue.

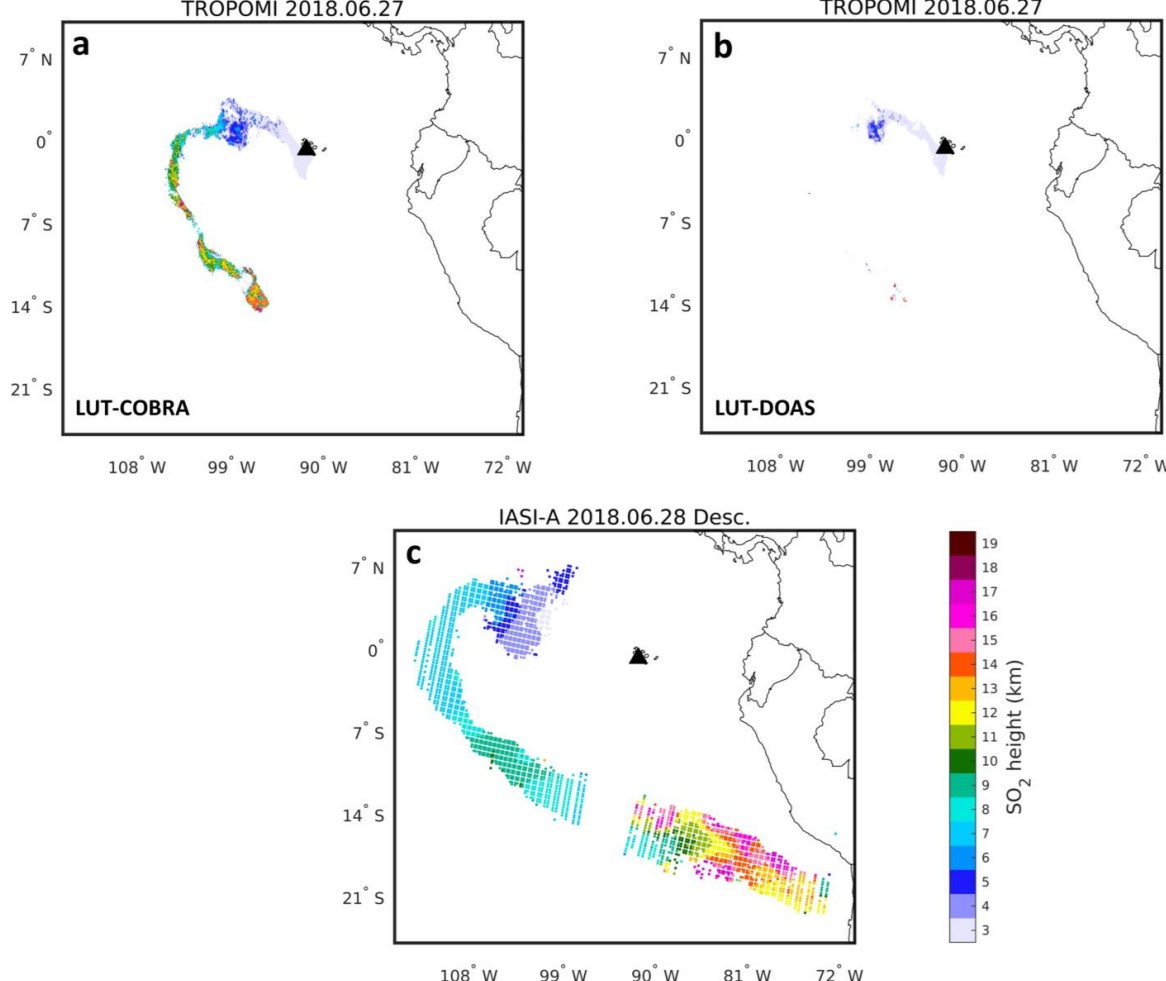

Figure 3: SO$_2$ layer height results for the eruption of Sierra Negra from (a) TROPOMI LUT-COBRA, (b) TROPOMI LUT-DOAS on 27 June 2018, and (c) IASI/MetOp-A on 28 June 2018 (descending orbit). The Sierra Negra volcano is marked by a black triangle.

Figure 4 a,c present two examples of SO$_2$ LH results for Raikoke on the 23 June 2019. Most of the SO$_2$ is found between 5 and 15 km, in agreement with Cai et al. (2021). The SO$_2$ distribution as a function of height seen in this example is also observed by other satellite data sets. The core of the plume is located in the 8-14 km altitude range similar as Hyman and Pavolonis (2020) and Hedelt et al. (2019), and is consistent with SO$_2$ profiles from the Microwave Limb Sounder (MLS).

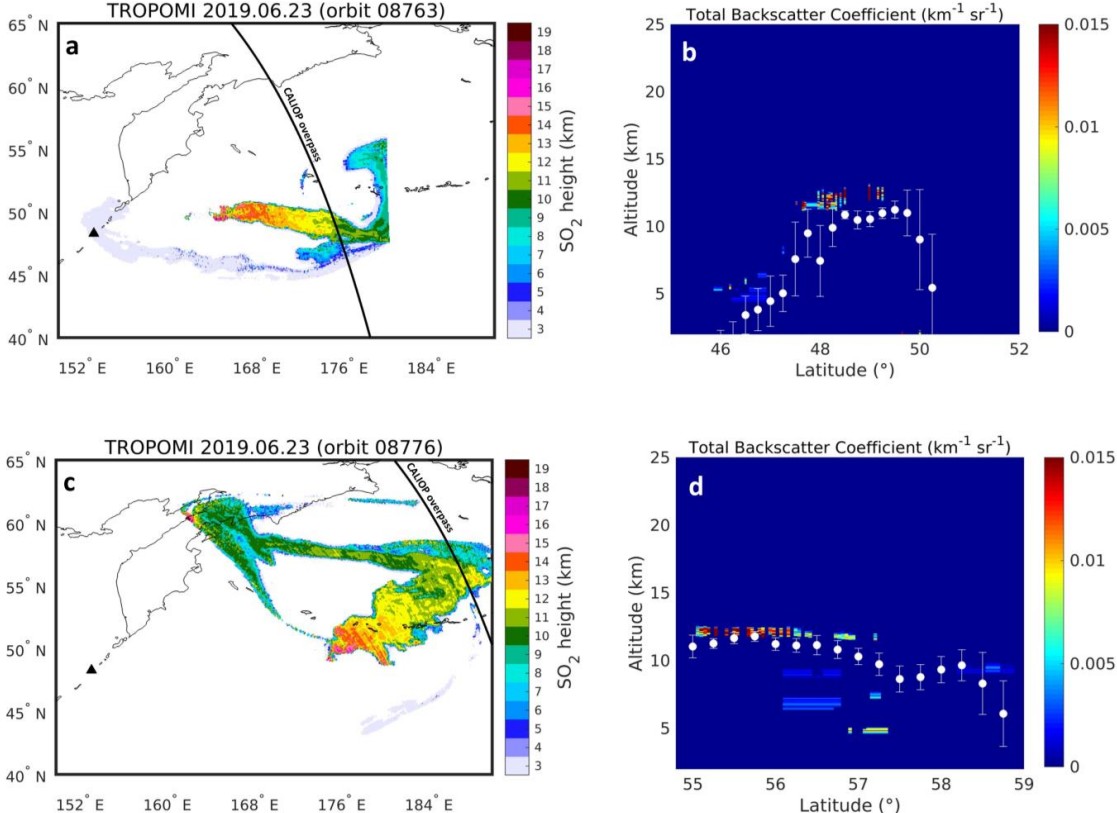

Figure 4: SO$_2$ layer height results from TROPOMI observations of the Raikoke plume on 23 June
2019, for orbits (a) 08763 and (c) 08776. The retrieved maximum SO$_2$ VCD is 613 DU and 274
DU, respectively. The Raikoke volcano is marked by a black triangle. The black lines indicate the
CALIOP overpasses within 1 hour. (b,d) Comparison between CALIOP plume detection from
measured 532 nm total backscatter coefficient and TROPOMI SO$_2$ LH results, for (a,c)
respectively. The TROPOMI values (in white) are the mean and standard deviation of the SO$_2$ LH
results within $0.25° \times 0.25°$ boxes, coincident with CALIOP.
Figure 4a,c also features a plume at much lower altitude, which is consistently observed in other
studies (e.g., Hyman and Pavolonis, 2020, de Leeuw et al., 2021, Muser et al., 2020). The
TROPOMI results of Figure 4 also agree reasonably well with IASI SO$_2$ heights. However, the
comparison is left out of this study, as it will be covered in a future publication (Vernier et al., in
preparation). Instead, we compare the TROPOMI SO$_2$ LH with nearly coincident CALIOP



observations of the Raikoke plume. Figure 4b,d show the comparison between the measured total
backscattered coefficient of CALIOP and collocated TROPOMI $SO_2$ LH, for the two orbits of Fig.
4a and c. Qualitatively, the TROPOMI $SO_2$ LH is in agreement with the aerosol features detected
by CALIOP. For instance, the plume at lower altitude (~46-48°N in Fig. 4b), is well captured by
TROPOMI. However, it is clear that overall the retrieved $SO_2$ heights are systematically lower
than CALIOP by 1-3 km. This finding is in line with the results of Koukouli et al. (2021) who
found similar low bias of the S-5P FP_ILM $SO_2$ height product compared to CALIOP. Note also
that the first TROPOMI observations of the Raikoke plume were made on 22 June 2019 (orbit
08749). For that plume, the algorithm retrieves $SO_2$ heights of ~8 km (not shown). Unfortunately,
there was no CALIOP measurement available on that day, but it appears that these heights are
much too low when compared to other data. This is not a problem specific to our algorithm, and it
highlights the difficulty to retrieve the $SO_2$ height in the UV for a scene with a mixture of $SO_2$ and
ash (e.g. Yang et al., 2010; Hedelt et al., 2019). Under these conditions, the LUT and LER
approximation fail to reproduce adequately the complex radiative transfer in the volcanic plume,
leading to a low bias on the $SO_2$ height, which can be as high as 5 km for fresh and thick ash
plumes.
A last test case is for the Ulawun volcano (5.05°S, 151.33°E, Papua New Guinea) that erupted
explosively on 26 June 2019 around 04:30 UTC and injected $SO_2$ at the tropical tropopause level
in the form of a well-defined umbrella cloud. On 27 June, TROPOMI (orbit 08821, approximate
time 04:00 UTC) observed a $SO_2$ plume over the region of Ulawun (Figs. 5a,b) with $SO_2$ LH
distributed mainly between 15 and 21 km (Fig. 5d), with a center-of-mass height of 17.7 km.
Conversely, the IASI/MetOp-A overpass at ~11:20 UTC on the same day revealed a plume of $SO_2$
injected in a narrower vertical layer, with center-of-mass height of 16.6 km, hence slightly lower
than the TROPOMI estimate. It is interesting to note that the total $SO_2$ mass inferred from
TROPOMI is in rather good agreement with the IASI estimate, within 10%. We argue that the
difference in the $SO_2$ mass distributions could actually relate to the limited sensitivity of the
TROPOMI retrievals for that plume. Indeed, the $SO_2$ columns from TROPOMI (Fig. 5b) are
modest, smaller than 20 DU for most pixels, and 9 DU on average. At this VCD level, the retrieval
error on TROPOMI $SO_2$ LH (Fig. 2, left panel) is significant for a $SO_2$ plume in the UTLS, around
1.5 km. This is compatible with the observed spread of the TROPOMI $SO_2$ mass distribution (Fig.
5d). Regarding the apparent ~1 km difference between TROPOMI and IASI $SO_2$ center-of-mass





heights, we can of course not completely rule out a systematic error on the IASI retrievals but it is
unlikely to explain fully the observed offset. Importantly, for these conditions (low to medium
VCD, high LH), the TROPOMI $SO_2$ height retrieval is exposed to forward model errors that could
easily explain a 1 km bias. For example, the LUT uses a simplified representation of the
atmosphere in terms of temperature and ozone profiles that can ultimately lead to systematic errors
in the Jacobians used for the retrievals.
Overall, the examples presented in this section show that there is a general good agreement
between TROPOMI $SO_2$ LH and other satellite heights estimates. Nevertheless, the results also
highlight limitations of the retrievals in some (difficult) conditions. More work would be needed
to improve these results.

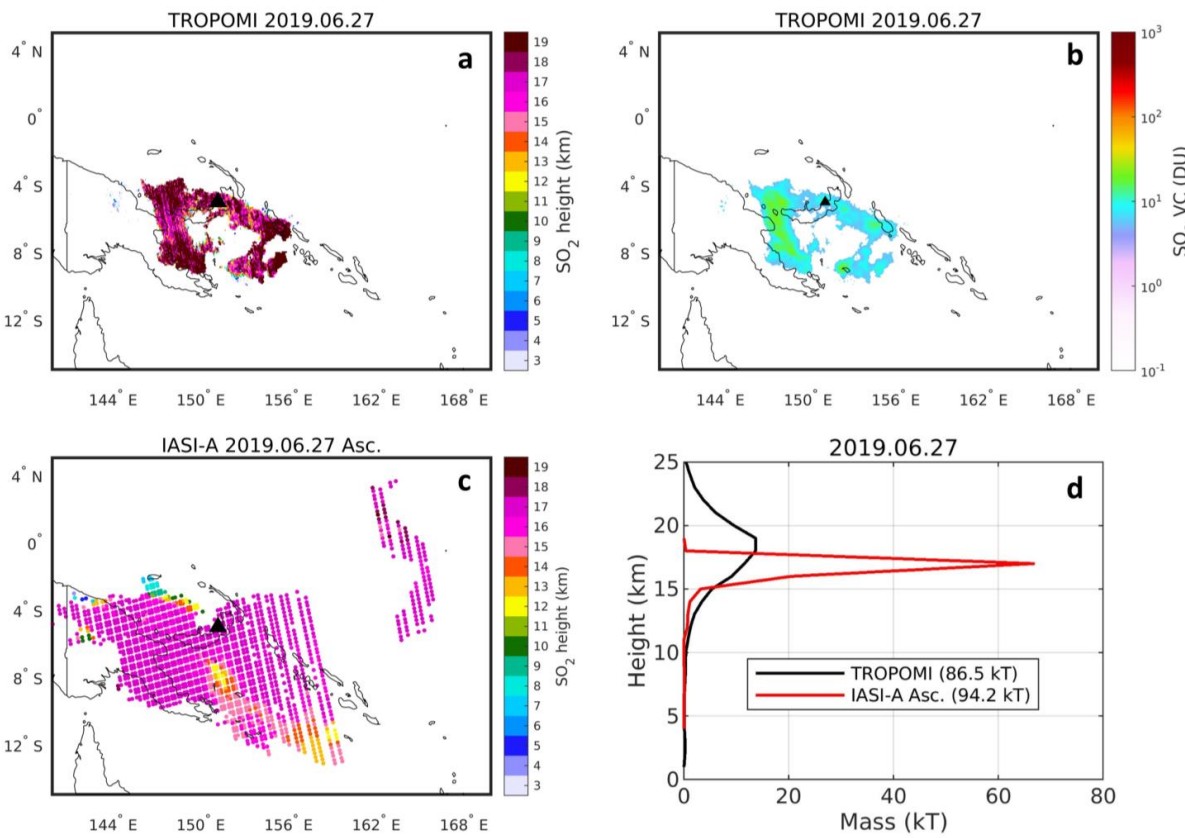

Figure 5: $SO_2$ plume on 27 June 2018 after the eruption of Ulawun. (a,b) TROPOMI retrievals of
$SO_2$ LH   and  VCD,  respectively,  (c)  $SO_2$  LH  from  IASI/MetOp-A  (ascending  orbit),  (d)



comparison of SO₂ mass histograms between TROPOMI and IASI. Total masses are indicated in
the legend. The Ulawun volcano is marked by a black triangle.
**3.2 Comparison with back trajectory analysis from PlumeTraj: the Etna case**
An independent validation of the SO₂ height retrievals of TROPOMI can be obtained from back
trajectory analysis. Recently, a general algorithm has been developed called PlumeTraj (Pardini et
al., 2017, 2018; Queiβer et al, 2019), which allows the height and age of volcanic SO₂ emissions
to be quantified for each TROPOMI pixel in a SO₂ image, using HYSPLIT back trajectories (Stein
et al., 2015). PlumeTraj leverages the fact that, because of the wind shear in the atmosphere, only
a limited range of back-trajectory altitudes connects a SO₂ pixel location with a given volcanic
vent. Importantly, when all pixels containing a volcanic plume are considered together, the height
and age parameters inferred by this method can be used in combination with the SO₂ column data
to reconstruct height- and time-resolved SO₂ emissions. This approach proves to be very powerful,
as it provides unique insights into the volcanic processes driving eruptions (Burton et al., 2021).
Here we have analyzed and compared the height results of TROPOMI SO₂ LH and PlumeTraj for
17 paroxysmal events of Mount Etna, Italy (37.75°N, 15°E), occurring in 2021.
Figure 6 presents an example of comparison, for a plume on February 19, 2021. It should be
stressed that the SO₂ plume heights, as shown in Fig. 6, are retrieved independently from each
other, as PlumeTraj only needs as input the observation time and pixel coordinates. For this event,
PlumeTraj derives SO₂ heights typically between 5 and 9 km with the highest values for the eastern
part of the plume and lower heights on the western part, and near the vent. Overall, this pattern is
well reproduced by our TROPOMI SO₂ LH retrievals - despite a few outliers. For the core of the
plume, the agreement between TROPOMI SO₂ LH and PlumeTraj is generally very good, with
differences mainly within ± 2 km. However, much larger differences are found for the edges of
the plume, where the TROPOMI SO₂ LH algorithm often retrieves SO₂ plume heights much lower
than PlumeTraj. We attribute this feature to an effect of the strong SO₂ horizontal inhomogeneity
within a TROPOMI pixel, which ultimately causes an underestimation of the retrieved SO₂ height
(consistent with Yang et al., 2010). This effect is also visible in Figure 4.

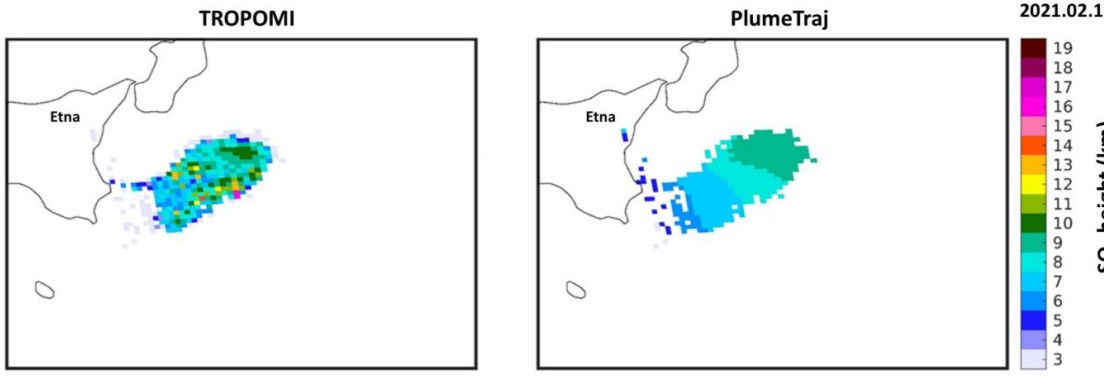

Figure 6: Example of SO$_2$ height results derived from TROPOMI and PlumeTraj for an Etna plume
on 19 February 2021.
Figure 7a summarizes the TROPOMI and PlumeTraj SO$_2$ height results, for all selected Etna
events. For each SO$_2$ plume, the center of mass height was calculated by averaging the SO$_2$ height
weighted by the SO$_2$ column amount in each pixel. For TROPOMI SO$_2$ LH, this is performed
using the retrieved SO$_2$ column data, while for PlumeTraj the SO$_2$ column is estimated using linear
interpolation of the TROPOMI operational SO$_2$ column product (given at 1, 7 and 15km; Theys et
al., 2017) to the altitude returned from the trajectory analysis. Note that all pixels with retrieved
SO$_2$ heights < 1 km were excluded from the analysis, in an effort to reduce the impact of the pixels
near the plume edges affected by pixel under-filling. As can be seen from Fig. 7a, TROPOMI and
PlumeTraj capture comparable SO$_2$ heights (in the range of ~ 4-11 km) and similar variability. The
estimated total SO$_2$ masses are also very consistent (not shown), and are in the range between 3.5
to 18.5 kt.


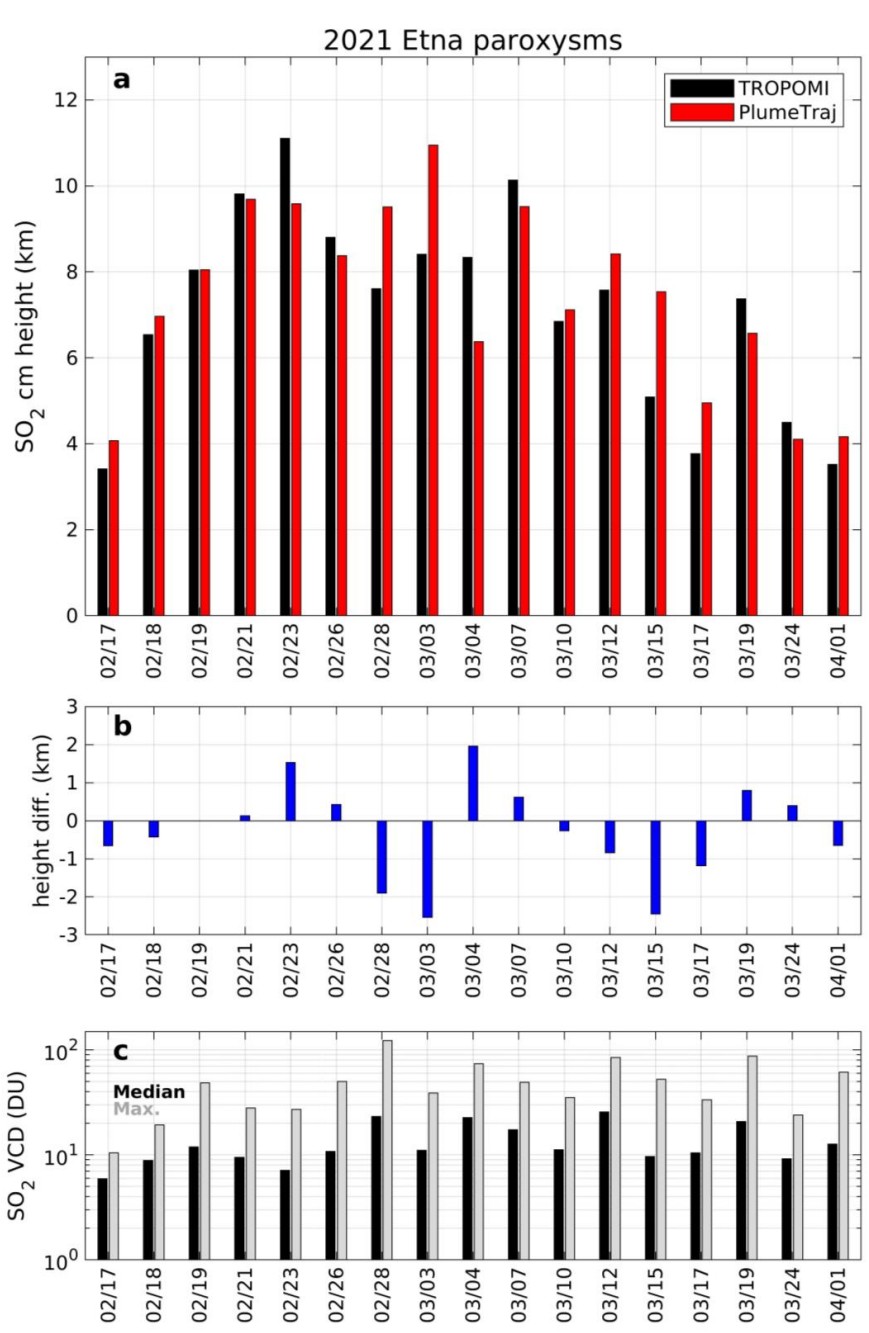



Figure 7: (a) Comparison of SO₂ center of mass height (km) from TROPOMI and PlumeTraj for
17 Etna paroxysmal events in 2021. (b) Differences of TROPOMI minus PlumeTraj SO₂ height.
(c) Median and maximum SO₂ columns retrieved by TROPOMI LUT-COBRA.
Figure 7b shows the differences in height between TROPOMI and PlumeTraj for all paroxysmal
events, and Fig. 7c summarizes the corresponding median and maximum SO₂ column values as
retrieved by TROPOMI SO₂ LUT-COBRA.
For about two third of the cases, the height estimates agree within ±1km. There are only a few
instances for which the height difference is higher than 1.5 km (in absolute value). However,
further investigation reveals that these cases correspond to particularly difficult conditions either
for the satellite retrievals (e.g., due to the presence of volcanic ash or because the plume was
narrow) or for PlumeTraj (unfavorable wind shear settings or because the plume was old).
Importantly, from Figs 7b and 7c, we cannot find a relation between the height discrepancy and
the SO₂ loadings. The median SO₂ VCD lies in the range between 6 and 25.6 DU, and Fig. 7
confirms that the TROPOMI SO₂ LH algorithm is able to derive reasonable SO₂ heights, even for
modest SO₂ vertical columns.
**3.3 Temporal analysis over degassing volcanoes**
Apart from eruptive events, TROPOMI is able to detect SO₂ emissions from degassing volcanoes
worldwide (Queiβer et al, 2019; Theys et al., 2019, 2021; Fioletov et al., 2020), and it is therefore
important to test our SO₂ LH algorithm on some of these volcanic emitters. Previous studies have
shown that IASI is sensitive to weaker volcanic emissions as well (Clarisse et al., 2012; Taylor et
al., 2018), and thus it is interesting to compare the TROPOMI and IASI height retrievals. For this,
we have considered two complete years of data (2020 and 2021) and analyzed time series of daily
height estimates from TROPOMI and IASI/MetOp-B over many different volcanic regions.
Because of the limited sensitivity of the satellites, it is clear that not all comparisons were
meaningful. However, for some volcanoes, the height of SO₂ was regularly retrieved by both
instruments over the studied period. Example of results are shown in Figure 8 for five active
volcanoes, namely Sabancaya, Peru (15.78°S, 71.85°W, summit elevation: 5967 m), Popocatepetl,
Mexico (19.02°N, 98.62°W, 5426 m), Tungurahua, Ecuador (1.47°S, 78.44°W, 5023 m),


Nyiragongo, DR Congo (1.52°S, 29.25°E, 3470 m), and Fagradalsfjall, Iceland (63.90°N, 22.27°E,
385 m). Note that the IASI height retrievals are the same as already introduced in Section 3.1,
except that additional criteria were applied to select the data with sufficient $SO_2$ signal and to make
the results comparable to TROPOMI. In particular, the same lower threshold of 5 DU for the
vertical column is applied.
As can be seen from Figure 8, a remarkable agreement is found between TROPOMI and IASI $SO_2$
heights. For Sabancaya, it is noticeable that successful $SO_2$ LH retrievals are frequent for both
instruments. The reason is likely due to the relatively high $SO_2$ columns there but also because
Sabancaya is an elevated site characterized by a dry atmosphere. To some extent, this is also true
for Popocatepetl and Tungurahua. On the contrary, a site such as Nyiragongo has a summit at
lower altitude and a wet atmosphere, resulting in fewer IASI retrievals. Finally, for Fagradalsfjall,
$SO_2$ was emitted much lower in the atmosphere (mainly below 2 km height) than for the other
cases. However, the match between TROPOMI and IASI is very good. In this case, IASI seems to
be able to retrieve $SO_2$ LH below 2 km. This is the result of dry conditions over Iceland for the
studied period. Note that the time series for Fagradalsfjall covers only a few months, after its 2021
fissure eruption.
Figure 8 indicates that TROPOMI tends to retrieve slightly lower $SO_2$ heights than IASI by ~ 0.5
km, although there is significant scatter in the height differences (standard deviation of about 1
km). The nature of this small systematic difference is unknown. However, this result nicely
demonstrates the value of the LUT-COBRA approach to infer the height of $SO_2$ for degassing
volcanoes or modest eruptions. For plumes in the lower troposphere, more frequent $SO_2$ heights
are retrieved with TROPOMI than IASI, which is an appealing aspect of the algorithm.





Figure 8: Time series of $SO_2$ height over five volcanic regions from TROPOMI and IASI/MetOp-B (daytime observations) for January 2020 to December 2021. Daily estimates of $SO_2$ center of mass height were calculated from quality-filtered data using fixed latitude-longitude boxes (from top to bottom) for Sabancaya (12-20°S, 70-78°W), Popocatepetl (15-22°N, 96.5-102°W), Tungurahua (5°S-1°N, 77-83°W), Nyiragongo (5°S-3°N, 25-32°E), and Fagradalsfjall (60-70°N, 10-32°W). For IASI, the same VCD lower threshold of 5 DU as TROPOMI is applied to select the data. The calculated $SO_2$ heights are shown only for days with at least 2 pixels. The mean and standard deviation of the differences between TROPOMI and IASI $SO_2$ heights are given as an inset for each plot.

## 4. **CONCLUSIONS**

We have presented a new algorithm to retrieve the $SO_2$ layer height and vertical column from TROPOMI UV observations. The retrieval scheme combines a large look-up-table to model the $SO_2$ signal and an error covariance-based approach, to represent the $SO_2$-free contribution of the spectrum. The method minimizes atmospheric or instrumental-related spectral interferences and reduces the $SO_2$ layer height error by a factor of 2 to 3 compared to a DOAS-fashioned implementation of the algorithm. This enables derivation of the $SO_2$ layer height with a precision better than 2 km for $SO_2$ columns as low as 5 DU and for a wide range of conditions. This is a significant improvement compared to other existing UV retrievals, which are limited to scenes of at least 20 DU of $SO_2$ columns.

We have demonstrated this approach on a number of eruptive events. Comparison with satellite IASI and CALIOP measurements and back-trajectory analyses indicate an agreement within 1-2 km, except for specific observations conditions. The presence of ash, in large amounts and at the same altitude as $SO_2$, causes the retrieval to underestimate the $SO_2$ height by several kilometers, in line with previous studies. Moreover, partially $SO_2$-filled scenes underestimate the $SO_2$ layer height, and this is mostly seen at plume edges. Despite these limitations, the performance of the algorithm is particularly good, especially for plumes below 10-12 km. We investigated the results against back trajectory analysis from the PlumeTraj toolkit, for relatively modest eruptions of Mount Etna in 2021. Using column-weighted average heights of $SO_2$, we found a very good agreement with PlumeTraj, even for total $SO_2$ masses of a few kt. Capitalizing on this, the temporal



evolution of TROPOMI $SO_2$ plume height was studied over some of the largest degassing
volcanoes, for a period of two years. An excellent correspondence is found between TROPOMI
and IASI with a mean difference of -0.5 km. This highlights the high sensitivity of the proposed
technique for the determination of plume height.
The algorithm is fast and could be adapted for near real-time implementation, and used e.g. in the
Support to Aviation Control Service, or other volcanic monitoring applications. The $SO_2$ height
results could also be helpful as a constraint for atmospheric dispersion modeling.
Future developments will focus on possibly enhancing the algorithm sensitivity, improving the
retrievals in the presence of aerosols, expanding the algorithm to stratospheric injection heights
(e.g., the 2022 eruption of Hunga Tonga–Hunga Haʹapai), and producing a proper quality
assurance flag.
**DATA AVAILABILITY**
The TROPOMI COBRA $SO_2$ layer height dataset is available from the corresponding author on
request. The ULB IASI $SO_2$ dataset is available from Dr. Lieven Clarisse on request. The output
of the PlumeTraj tool is available from Dr. Mike Burton on request. NASA CALIPSO data can be
downloaded from https://www-calipso.larc.nasa.gov/ (last access: 30 March 2022).
**AUTHOR CONTRIBUTIONS**
N.T. prepared the manuscript and figures with contributions from all the coauthors. N.T., C.L.,
J.v.G., H.B., I.D.S., M.V.R. contributed to the development of the LUT-COBRA, processing of
the data and satellite comparison. L.C. analyzed and provided IASI data. M.B. and M.V. analyzed
and provided PlumeTraj data. All authors contributed to the interpretation of the results and
improvement of the manuscript.
**COMPETING INTERESTS**
The authors declare that they have no conflict of interest.



## 1 ACKNOWLEDGEMENTS AND FINANCIAL SUPPORT

We acknowledge financial support from ESA Level-2 Prototype Processor of the future
Copernicus Sentinel-5 satellite (contract #4000118463/16/NL/AI), ESA S5P MPC
(4000117151/16/I-LG), Belgium Prodex TRACE-S5P (PEA 4000105598), Horizon 2020
EUNADICS-AV (grant agreement no. 723986), SESAR H2020 ALARM and the OPAS Engage-
KTN (grant agreement no. 783287) projects. The ALARM project has received funding from the
SESAR Joint Undertaking (JU) under grant agreement No 891467. The JU receives support from the
European Union's Horizon 2020 research and innovation programme and the SESAR JU members other
than the Union. We thank EU/ESA/KNMI/DLR for providing the TROPOMI/S5P Level 1 and Level-2
products. This paper contains modified Copernicus data (2018/2020) processed by BIRA-IASB.
L. C. is a research associate supported by the Belgian F.R.S-FNRS.

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



1   **Appendix**

2   **Table A1.** Input settings used to generate radiative transfer simulation.

| | |
|---|---|
| **Model** | Lidort RRS; Raman scattering switched off. |
| **Wavelength range** | 309-329 nm, 0.05 nm spectral sampling. All spectroscopic data are pre-convolved using 0.05 nm box-car function. |
| **Solar spectrum** | Chance and Kurucz (2010). |
| **Cross-sections** | Ozone: Serdyuchencko et al. (2014), $SO_2$: Bogumil et al. (2003)<br>Temperature dependence of the cross-sections are accounted for in the simulations. |
| **Atmosphere** | Ozone and temperature profiles: total ozone column classified profiles from Lamsal et al. (2004). All available climatological profiles are averaged for each total ozone column value of 145, 175, 205, 235, 295, 355, 415, 475, 535 DU.<br>Pressure profile (US Standard). |
| **$SO_2$ profiles** | Gaussian profiles with full width at half maximum of 500 m, peaking at $SO_2$ height and scaled to VCD as in Table 1. |
| **Aerosols and clouds** | Not included in the simulations (treated as LER by the algorithm). |
| **Output** | Radiance and $SO_2$ slant optical depth (log ratio of radiance simulations including/not-including $SO_2$ absorption). |

