# Peer review of "Improved retrieval of SO2 plume height from TROPOMI"

_Atmospheric Measurement Techniques, 2022_

## Author Comment (AC1)

**Replies to reviewers of amt-2022-148**

The authors wish to thank all reviewers for their time and constructive comments that contributed to improve the manuscript. A point-by-point reply to all reviewer comments can be found below. Changes in the revised document are in blue.

**Anonymous Reviewer #1**

**Review of the manuscript "Improved retrieval of SO2 plume height from TROPOMI using an iterative Covariance-Based Retrieval Algorithm" by Theys et al.**

The manuscript presents a new and appealing method to retrieve the SO2 plume height from TROPOMI UV measurements even for low SO2 VCDs. The paper is very well written and I suggest publication after minor revision.

The suggested covariance-based method is an extension of the existing COBRA algorithm that has been developed by the authors to retrieve the SO2 SCD from TROPOMI measurements and uses a method that was so far only used for IASI SO2 retrievals. It allows for the retrieval of SO2 LH even for SO2 VCDs as low as 5DU with a precision better than 2km.

I have only a few minor points, which I would like the authors to address in a revision of the manuscript:

Sect. 2.1:

- You are using the Bogumil et al 2003 SO2 cross-section for LIDORT. Have you considered using improved SO2 XS by Birk et al from the ESA IAS SEOM project?

  Yes, we have considered to use this new spectroscopic database. However, the report from the ESA IAS SEOM project indicates a rather small impact of this new spectroscopy on the retrieved SO2 columns, typically less than 10% for the wavelength range used here. Therefore we were not convinced of the necessity to regenerate the LUT. For further work, it would be good though to thoroughly assess the impact on the SO2 plume height results.

- Since you apply an I0 correction to the SO2 cross-section, please mention which value you have used. Could you give numbers of how strong the effect is when (not) using the I0 correction? Does it have an effect for low SO2 VCDs?

  In theory, the Io correction applies to the slant optical depth. It is for DOAS type of applications that the Io correction is applied to the cross-section (generally by assuming a representative SCD). Here we simply use: $OD_{corr}$ = -log [ SF x (Io.exp(-OD)) / SF x Io] where Io is the high resoltion solar spectrum (Kurucz) and 'SF x ..' stands for slit function convolution. We have added this information in the text. The effect of this correction on the $SO_2$ optical depth for large columns (>~50 DU) can be of several $10^{-3}$, which is not

negligible and higher than the noise level. The corresponding impact on the retrieved SO₂ height is hard to quantify and will depend on the conditions.

- Since you are using LIDORT to calculate SO2 OD spectra, why do you also take into account O3 absorption? Maybe I am missing something here, but you are only interested in SO2 OD spectra itself, right?

  The SO2 spectra is a 'slant OD' and therefore accounts for possible influences on the SO2 air mass factor. In other words, the photon path depends on SO2 itself (of course) but also on other parameters (as listed in Table A1), O3 absorption being one of them. We have clarified this point in the text.

Table 1:

- Is there a reason why you have chosen a coarser SO2 LH grid at high altitude for generating your LUT? What is the effect when using a finer grid?

  The main reason is to reduce the number of simulations to generate the LUT, knowing that the scattering weights are essentially independent on the altitude above ~15km. In next algorithm versions, a finer grid could be used but it is not expected to change the results significantly.

- Is there a reason why you only extend the LH grid until 25km and not higher, e.g. up to 30km?

  No, in principle it could be extended higher for next versions of the algorithm.

- At the nadir point of the TROPOMI swath, the RAA shows a strong jump by up to 90° from one pixel to the neighboring pixel along the scanline. Does this cause problems/jumps in the retrieved SO2 LH/VCD between pixels in the center of the swath since the OD is interpolated at strongly different RAAs?

  We have checked the RAA dependence for a number of volcanic plumes extending over a large portion of TROPOMI but we could not identify any jump in the data. Our understanding of this is that the RAA dependence of the measurement sensitivity is small in particular close to nadir so that if a jump exists in the retrieved height it is likely small compared to other sources of uncertainties (or real variability).

Sect 2.2: When you construct the covariance matrix S, how is the whole retrieval affected by clouds? In detail, when the SO2-free spectra contain a huge number of cloudy pixels, whereas the retrieved SO2 pixels are cloud-free (or vice-versa), is the covariance matrix still well-posed?

The same applies to ash clouds in the SO2-free pixels used to construct S. For these cases I would like to see some more discussion on the effect of the SO2 LH/VCD retrieval, if possible

This is a good point. For the construction of S, the selection of SO2-free spectra covers a region of ~1600km (along-track) around the pixel of interest. Doing so permits to sample a wide range of cloudiness. Therefore the cloud condition of the pixel of interest is likely represented by the covariance, at least to some degree. For ash clouds, the situation is probably different because the reference set of spectra is not covering well these conditions of low UV intensities. In addition, and as stated in the initial manuscript, the LUT it-self is not designed for ash plumes. Consequently, the accuracy of the retrieved SO2 plume heights is limited for such conditions. We have added a discussion on the representativeness of S and the effect on the retrievals.

Technical corrections:

Sect. 3.1, Acknowlegdements and Reference sections show some strange bold uppercase 'K' letters (at least in my pdf viewer). Please correct

These are not appearing on my screen.

Reference to Koukouli et al 2021 is officially published: Atmos. Chem. Phys., 22, 5665–5683, 2022, https://doi.org/10.5194/acp-22-5665-2022. Please update the reference

This is corrected in the revised manuscript.

**Anonymous Reviewer #1**

The manuscript of Theys et al. 2022 "Improved retrieval of $SO_2$ plume height from TROPOMI using an iterative Covariance-Based Retrieval Algorithm" presents an interesting new retrieval approach to retrieve both SO2 VCD and LH based on UV Earthshine spectra.

Although the paper is clearly written, I am not sure to understand how the retrieval itself works:

In Section 2, equation 3 you write that $x_i$ is the state vector representing $VCD_i$ and $LH_i$ and $k_i$ contain the Jacobians. So is $k_i$ in the end a 2d vector containing the two Jacobians as a function of wavelength that you multiply with your covariance matrix S?

Yes, it is correct that the forward model is composed of two Jacobians vectors which multiply the inverse of the covariance matrix. This is described in Section 2, Eq. 3.

How dependent are your results on the apriori $VCD_0$ and LH @7km? I.e. what happens if this initial estimate is completely wrong and the plume is at 20km in reality (with much lower true $VCD_0$)?

We have made these tests and the conclusion is that the retrievals are essentially insensitive to the prior VCD and LH. This is clarified in the revised text.

In your construction of the covariance matrix S you write that you follow the approach of your Theys 2021 paper and filter out pixels with VCD > 2.5 x VCD SNR? Do you also put an upper limit of the LH SNR to filter out pixels?

No, for the construction of S only the column amount is use for the filtering.

**Anonymous Reviewer #2**

The study ``Improved retrieval of SO2 plume height from TROPOMI using an iterative Covariance-Based Retrieval Algorithm'' by Theys et al. introduces an optimal estimation retrieval algorithm to retrieve $SO_2$ column and layer height of volcanic injections from nadir measurements between 309 and 329 nm by TROPOMI. First, the retrieval is described and its theoretical uncertainties are assessed. Then examples of the retrieval for various volcanic plumes are presented. The layer heights are verified by comparison with IASI measurements and reconstructed plume heights from a backward trajectory approach. In general I found this paper well structured and well written. Hence I recommend this paper for publication after addressing the minor comments below.

**General comments:**

This paper contains many abbreviations, which often hamper the reading flow (e.g. ``The LER albedo is retrieved by matching the measured mean radiance to a LUT of radiances (generated in parallel to the SO2 OD LUT), and which depends on SZA, VZA, RAA, surface height and albedo, with the same grid definition as in Table 1.``). I suggest using less abbreviations, i.e. write out LH, LER, LUT, OD, VCD.

We have checked if all abbreviations are defined in the text. We prefer to keep the number of abbreviations as they are, to limit the length of the paper. The use of abbreviations is normal for such a technical paper. However, at some places in the text, we agree that the use of abbreviations is not ideal and we have slightly adapted the text accordingly.

Please note, usually it is called a volcanic ``injection'' if a volcanic eruption reaches the upper troposphere and stratosphere. Volcanic ``emission'' is rather used for low altitude degassing. Please consider replacing ``emit, emission,...'' by ``inject, injection,...'' (e.g. p2ll6,9,11).

Wherever possible, we have changed the text according to this suggestion.

For the setup of the LUTs the US standard atmosphere was used. This is rather representative of midlatitude conditions. Could this have a (negative) impact on the retrievals of plume heights close to the tropical tropopause? Did you perform sensitivity tests for tropical, midlatitude, and polar atmospheric conditions?

Please note that the US standard atmosphere is used only for the pressure profile. The main reason is that the use of a single pressure profile is practically convenient for the pressure-height conversion throughout the algorithm. However, a certain dependence of the temperature vertical distribution with latitude is considered through the use of the Lamsal et al climatology (Table A1). We are aware that our representation of the atmosphere is a simplification but it is also not compromising the performance of the retrievals either. Importantly the air density profile (which drives the scattering and hence the sensitivity) is not varying dramatically from the tropics to the poles (std less than 10% of the mean profile).

**Specific comments:**

p2l4: Please specify what ``difficult observation conditions'' means.

This is clarified in the revised manuscript.

p6l6: Please explain ``Lambertian Equivalent Reflectivity'' or add a reference.

This is clarified in the revised manuscript.

p6l7: ``More details are given below.'' Please specify details on what are given? Simulation setup? Simulation output parameters?

This is clarified in the revised manuscript.

p6l13: Please add one sentence describing the ``so-called solar $I_0$ effect''.

This is clarified in the revised manuscript.

p6l14-15: It is not clear to me if 450 refers to the across-track pixel position or to the ISRF parameters.

It stands for the across-track pixel positions. This is clarified in the revised manuscript.

p6l30: Why does the LER approach work only well in principle? How is it in practice? Please provide a reference.

This is clarified in the revised manuscript.

p8l21: Is there a default value for VCD0? Which one?

This is detailed in the first paragraph of page 9.

p9l9: Is minimum height + 1 km = surface height + 1 km?

Yes. This is clarified in the revised manuscript.

p9l10: Is maximum height - 1 km = 24 km?

Yes. This is clarified in the revised manuscript.

p9l18: How much is 300 pixels roughly in km?

About 1600km. This is clarified in the revised manuscript.

p9l20-22: Is there an upper limit in terms of distance to the central pixel?

No.

p10: How did you predefine the Jacobians and the covariance matrix?

For this section, the Jacobians are taken from the LUT, for the conditions considered for this exercise (cf caption of Figure 2). The covariance matrix is based on TROPOMI measurements extracted for conditions typically close to the ones considered in Figure 2.

Figure 2: Is this the ``sweet spot'' configuration where the layer height error is smallest for low DU? For which surface type (i.e. ocean, land, ice) is an albedo of 5% representative? In the text you mention that you tested the impact of the albedo and SZA. What about the impact of surface height, ozone, RAA and VZA?

An albedo of 5% is typical of land surface, free of snow and clouds. It is not clear what the reviewer means by "sweet spot". Anyway, we always observe a better performance of LUT-COBRA compared to LUT-DOAS, for the different scenarios we have looked at. A systematic investigation of the performance of the algorithm for all conditions is difficult. Here we simply intended to illustrate the sensitivity of the retrievals for one particular case. The important thing is that the error characterization can be done for each pixel and is actually reported in the product output file.

p13l22: What are S-5P FP_ILM results?

These are the results from the Inverse Learning Machine scheme of Hedelt et al. (2019). This is clarified in the revised manuscript.

p13l23-24: How do you know from TROPOMI data that the plume had multiple layers?

Here we mean multiple heights. This is clarified in the revised manuscript.

Fig. 3: Could you please add contours to DU>20 to all panels or show contours for DU>5 to the DOAS panel? I see the point that you want to compare the layer heights, but I find the different shapes of the plumes confusing. If I understood correctly the DOAS method can retrieve lower DU, but has a larger uncertainty for <20 DU. Also, please add the orbit footprint to make the gap between two orbits better visible, especially in the region around the volcano.

This has been changed in the revised manuscript.

Fig. 4: Please add the orbit footprint here too. Also I think it makes sense to cut Fig b and d at 20 km, as the color bar in a and b is also cut at 20 km.

This has been changed in the revised manuscript.

p17l7: Could it be that the larger difference to CALIOP aerosol height is due to the fact that for CALIOP the aerosol layer top altitude is compared to the TROPOMI layer height that is more representative of the center of the $SO_2$ layer?

Yes, this can be a part of the explanation. We have added one sentence on this in the revised manuscript, complementing the already existing discussion on the effect of aerosols on the retrievals.

Fig. 5: See general comment on using only the US standard atmosphere instead of representative profiles for the tropics and midlatitudes. How well do the TROPOMI and IASI results compare to CALIPSO aerosol heights?

Please read above the comment on the use of US standard atmosphere. CALIOP measurements on that day are not very conclusive.

p18l9: Please specify ``difficult conditions''.

This is clarified in the revised manuscript.

p19l10-15: The method of using backward trajectories from satellite observations of plumes back to the volcano has been also used in other studies, e.g. Wu et al., 2017, Cai et al. 2022.

We have cited the study of Wu et al., 2017 in the text.

p19l20: Does PlumeTraj only rely on position and time, or does it also consider the DU at each pixel to weight the reconstructed heights (e.g. high weight for large DU, low weight for small DU)?

Plumetraj is considering the column amount in each pixel to reconstruct the height-time resolved SO2 emissions. This is explained in the text and the related references.

p19l25-29: Could this difference in plume height at the plume edges be also due to low DUs at the edges? How can you tell there is no underlying low altitude plume? Which altitude range is considered by PlumeTraj? Would this ``edge-effect'' still be there if you used e.g. 10 DU as a selection criterion for the layer height retrieval?

This edge effect is observed not only for low DUs. Obviously, at plume edges it is easy to find pixels with low columns but for this particular example there are also SO2 VCDs of ~30 DU or more that show the same behavior. In fact for the Raikoke example, the edge effect is also clear (e.g. Fig 4.a), despite the very high columns. The problem arises when a pixel is party filled with SO2. The measured intensity is then the sum of an SO2-free part and a contribution including an SO2 absorption, which is non-linear. This creates a spectral distortion that is not accounted for in the forward model. This effect has been reported already by Yang et al. (2010).

The PlumeTraj tool considers altitude plumes from the surface to the lower stratosphere. Should there be an underlying low altitude plume, it is likely that PlumeTraj would capture it at least to some degree. Note that because of the vertical wind shear, it is unlikely that a near-surface plume would be transported exactly the same way as a plume at 6-9km height as shown in this example.

Fig. 6: According to Fig. 7 Fig. 6 shows the best case. What does a bad case, e.g. 3 March or 4 March look like? Is there also an ``edge-effect'' visible?

Yes, the edge-effect is also observed. The initial text discusses the reasons for possible discrepancies. Therefore we think the text does not have be changed.

 p22l12: How old is ``old''? Days, weeks?

Typically it is a ~1day old. This is clarified in the revised manuscript.

Fig. 8: The figure quality needs to be improved. I don't understand why there are lines for TROPOMI.

The figure is improved in the revised manuscript.

p25l8-10: How were the deviations calculated? On a daily basis? There are lines for TROPOMI and points for IASI data. Sometimes there are data points for IASI, but not TROPOMI data.

The standard deviations are calculated for the complete period based on the SO2 height daily estimates. The figure has been improved.

**Technical comments:** -> all technical comments have been addressed in the revised manuscript.

p2l4: expect -> except

p3l27: remove ``obviously''

p5l16: $x_l$ -> $x_i$

p6l28: aerosols layer -> aerosol layers

p7l2: aerosols -> aerosol particles

p7l5: Please introduce the abbreviation OCRA/ROCINN CRB.

p7l9: Please introduce the abbreviation RAA.

p13l26: Please introduce the abbreviation CrIS. This is introduced already in section 1.

p13l31: sensors -> measurements

p17l20: observed a -> observed an

Fig. 5 caption: 2018 -> 2019

p22l29: Example -> Examples

**References:**

Wu et al., 2017; https://doi.org/10.5194/acp-17-13439-2017

Cai et al. 2022; https://doi.org/10.5194/acp-22-6787-2022

**Anonymous Reviewer #3**

This paper describes a new algorithm using TROPOMI UV measurements to retrieve effective volcanic SO2 height. This is an extension of the COBRA SO2 algorithm previously described by the authors, with the addition of SO2 height (in addition to SO2 VCD) to the retrieved state vector. The authors presented several examples of the TROPOMI SO2 height retrievals, including both degassing volcanoes and large explosive eruptions (Raikoke 2019). Comparisons with thermal IR SO2 height retrievals (IASI), CALIPSO lidar measurements, and trajectory-based height estimates show generally good agreement between TROPOMI retrieved SO2 heights and those datasets, even for relatively small SO2 loading. Overall, the paper is well written and the results are quite impressive for a rather challenging problem. The SO2 height retrievals also can have a host of different applications and should be of interest to the readers of AMT. I'd recommend that the paper be accepted for publication after minor revisions.

Specific comments

My main concern is that the nodes in the LUT (Table 2) appears to be relatively coarse especially for larger SO2 amounts. This may lead to interpolation errors at shorter wavelengths. Can the authors compare the interpolated Jacobians with model calculated ones to assess the uncertainty from interpolation?

We understand the concern about the LUT in particular for the SO2 column grid and we have added a discussion on this in the revised text. We have made a comparison of SO2 optical depth spectra for a forward simulation considering an SO2 VCD of 150 DU (i.e. in between the 2 nodes at 100 and 200 DU) and the interpolated data. The results are shown on the figure below. The other parameters for the simulations are: SO2 height=7km, SZA=30°, VZA=RAA=0°, Albedo=5%, TO3=295DU, surface height=0km.

[Figure]

As can be seen, the results are very similar, differing by less than $10^{-3}$ hence close to the noise level. This means the LUT approach is able to reproduce the SO2 signal to a large extend. There might be some (extreme) conditions for which the interpolation errors are larger, but these will likely be small in comparison to other sources or errors (e.g. aerosols).

Page 9, lines 8-12: would an out-of-bound value in VCD or SO2 height generally lead to failed retrieval?

Yes, this can happen but it is not necessarily the case. The highest SO2 VCD of the LUT is 1000 DU, which is a considerable high value and is suitable for most cases. For high SO2 loadings, what can happen for the first or second iteration is that the retrieval returns an SO2 height too low and hence a column too high but it generally stabilizes after few iterations. If not, the maximum number of iterations is reached, and the retrieval is indeed reported as failed. However, this is an exception.

Page 9, lines 22-23 - is the SZA limit also applied to the selection of pixels to construct the covariance matrix?

Yes. This is clarified in the revised manuscript.

Page 10, line 22: "referred as" should be "referred to as".

This is changed in the revised manuscript.

Figure 3: there appears to be more structure in Figure 3a LUT-COBRA results - is there an explanation for this?

Compared to IASI, the results of TROPOMI are more structured indeed, in part due to the improved spatial resolution of TROPOMI. It is difficult to prove these structures are all real but there are also explanations why IASI looks smoother. First, IASI returns integer values for the heights while TROPOMI results are rounded and this introduces some variability. Second, the IASI retrievals are averaged over the neighboring pixels to remove 'salt-and-pepper' noise, and this smoothed possibly some real variability as well.

Page 17, line 15: suggest changing "as high as" to "as large as" to avoid confusion.

This is changed in the revised manuscript.

Page 19, lines 27-29: there appears to be some striping in Figure 4 even for pixels that are near the center of the plume. Could this have anything to do with the initial conditions based on the operational SO2 product?

There are no stripes visible in the operational SO2 column product, therefore these are not related to the initial conditions. One should keep in mind that each row is essentially a different instrument and the retrieval is performed separately for each row. The striping related difference from one row to the next is quite small (<1km) and there are many sources of uncertainty that could lead to such feature.

Figure 7: it would be useful to have some statistics of the comparison (e.g., r, RMSE).

These are added in the revised manuscript.